# A simple pressure-assisted method for MicroED specimen preparation

Jingjing Zhao [1], Hongyi Xu [1✉], Hugo Lebrette[2], Marta Carroni [2,3], Helena Taberman[4,5], Martin Högbom[2] & Xiaodong Zou [1✉]

Micro-crystal electron diffraction (MicroED) has shown great potential for structure determination of macromolecular crystals too small for X-ray diffraction. However, specimen preparation remains a major bottleneck. Here, we report a simple method for preparing MicroED specimens, named Preassis, in which excess liquid is removed through an EM grid with the assistance of pressure. We show the ice thicknesses can be controlled by tuning the pressure in combination with EM grids with appropriate carbon hole sizes. Importantly, Preassis can handle a wide range of protein crystals grown in various buffer conditions including those with high viscosity, as well as samples with low crystal concentrations. Preassis is a simple and universal method for MicroED specimen preparation, and will significantly broaden the applications of MicroED.

[1] Department of Materials and Environmental Chemistry, Stockholm University, Stockholm, Sweden. [2] Department of Biochemistry and Biophysics, Stockholm University, Stockholm, Sweden. [3] Science for Life Laboratory, Stockholm University, Solna, Sweden. [4] Max Delbrück Centrum for Molecular Medicine, Berlin, Germany. [5] Macromolecular Crystallography, Helmholtz-Zentrum Berlin, Berlin, Germany. ✉email: hongyi.xu@mmk.su.se; xzou@mmk.su.se

Three-dimensional electron diffraction, known as microcrystal electron diffraction (MicroED)[1,2], has shown great potential for structure determination of macromolecules from nano- and micron-sized crystals, which are too small for X-ray diffraction[3-5]. This method has attracted large interest and extensive attention in structural biology, structural chemistry, and the pharmaceutical industry during the past years. However, until now, only a few macromolecular structures solved by MicroED are reported in the PDB database and all except R2lox[3] had been previously determined by X-ray diffraction. The R2lox structure[3] was solved by molecular replacement using a search model of 35% sequence identity (PDB code 6QRZ). MicroED data collection is usually very fast (1–3 min)[2,6,7] by applying continuous rotation of crystals and a hybrid pixel detector or complementary metal-oxide semiconductor detector in movie mode. Data processing and structure determination can be done using standard X-ray crystallographic software suites. Despite those advances in MicroED experiments, the major bottleneck has been specimen preparation, which is most delicate and time-consuming.

In MicroED experiments, sub-micrometre thick crystals are needed in order to allow the electron beam to penetrate through the specimen and minimize multiple scattering. The surrounding ice also needs to be as thin as possible to improve the signal-to-noise ratio while still protecting the protein crystals from dehydration. Suitable crystal size can be obtained by adjusting the crystallization conditions[8,9], by segmenting large crystals using mechanical forces (e.g., vigorous pipetting, sonication, vortexing with beads)[10], or by focused ion beam milling under cryogenic conditions (cryo-FIB)[11-14]. High molecular weight polymers, like polyethylene glycols (PEG), are common and popular agents to produce volume-exclusion effects for successful protein crystallization[8,9,15]. However, their addition makes the buffer viscous. Lipid cubic phase (LCP) is commonly used for crystallization of membrane protein, and they are extremely viscous as toothpaste[13,16,17]. It has been very challenging to prepare MicroED samples for crystals grown in viscous buffers[4,13,18]. The pipetting-blotting-plunging routine[19], originally designed for single-particle cryo-EM specimen preparation, has two major drawbacks for MicroED specimen preparation; (1) many microcrystals are removed by blotting and (2) it is insufficient in removing viscous liquids[18]. Manual back-side blotting[18], direct crystallization on EM grids[20], and cryo-FIB[13,14,17,21] have been proposed as possible solutions to deal with the above problems. However, a simple and universal method for MicroED specimen preparation is still missing. There is an urgent need to develop such a method in order to apply MicroED on a wide range of macromolecular crystals.

Here, we describe a pressure-assisted method, named Preassis, for the preparation of MicroED specimens. The excess liquid is removed through the EM grid with the assistance of pressure. Preassis is applicable for a wide range of protein crystal suspensions with both low and high viscosities. It can preserve up to two orders of magnitude more crystals on the TEM grid compared with Vitrobot, which has a unique advantage to study samples with low crystal concentrations. The ice thicknesses can be controlled by tuning the pressure in combination with appropriate carbon hole sizes of EM grids. We provide detailed experimental guidance in finding appropriate parameters for preparing MicroED specimens of new protein crystal samples. More importantly, the Preassis method is simple and easy to implement, making it widely accessible to cryo-EM labs at a very low cost.

## Results and discussions

The basic concept of Preassis is to pull a portion of sample suspension through an EM grid and simultaneously remove excess liquid from the backside of the grid with the assistance of suction/pressure. In its simplest setup (Fig. 1 and Supplementary Fig. 1), an EM grid is placed on a filter paper which is rested on the mouth of a Buchner flask and pumped with a certain pumping speed. A droplet of the sample suspension is then deposited onto the grid. Due to the suction underneath the filter paper, the excess liquid is "pulled" through the grid. Then the grid is manually picked up using a tweezer and plunged into liquid ethane. Details of this setup and specimen preparation procedures are available in Supplementary Methods. The overall vitrified ice thickness on the EM grid can be tuned by changing the pressure, carbon hole size of the EM grid, and the time over which the pressure is applied. While the pressure can be changed continuously, the change of the carbon hole size is done by choosing the type of holey carbon EM grids. In the current setup, the pressure is adjusted by changing the pumping speed, where the pressure is linearly proportional to the pumping speed in the range of 20–80% (Supplementary Fig. 2). It is worth mentioning that the ice layer thickness can vary throughout an EM grid prepared by the current Preassis setup, as shown in Supplementary Fig. 3. This may be due to non-uniform contact of the EM grid to the filter paper. Such a gradient of ice thickness may not be a disadvantage, and instead, it increases the chance of finding suitable crystals for MicroED data collection.

We compared EM specimens prepared by the double-side blotting in Vitrobot and Preassis, by studying the crystal density, MicroED data quality, and ice thickness. We also studied the influence of humidity on ice thickness. We chose a non-viscous suspension of tetragonal lysozyme microcrystals with a high crystal concentration for the study. For each method Preassis or Vitrobot, at least three copies of grids were prepared using the same condition. We found Preassis requires much lower crystal concentrations than Vitrobot, as shown in Fig. 2 and Supplementary Figs. 4 and 5. When a crystal suspension with low concentration (i.e., diluted by 500×) was used, the grids prepared by Preassis had thousands of microcrystals (Fig. 2a–c and Supplementary Figs. 4a–c and 5a–c), while those prepared by Vitrobot were almost empty (Fig. 2d–f and Supplementary Fig. 4d–f). To reach a sufficient crystal coverage for MicroED experiments, a crystal suspension with at least 250 times higher concentration (i.e., diluted by only 2×) was needed for Vitrobot, as shown in Fig. 2g–i and Supplementary Fig. 5d–f. This comparison shows that Preassis can keep around two orders of magnitude more crystals on the EM grid than Vitrobot. This is mainly because, by Preassis, excess liquid is removed through the carbon holes of an EM grid where crystals are kept by the holey carbon film. In the case of Vitrobot, most excess liquid was taken away from the front side of the grid by a filter paper where

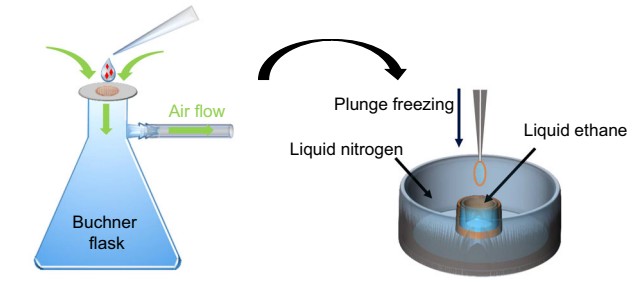

**Fig. 1 Schematic drawing of the Preassis setup.** A drop of the microcrystal suspension is transferred onto an EM grid resting on a support (e.g., filter paper). Simultaneously, the extra liquid is removed through the support with the assistance of pressure, and followed by manually plunge freezing in liquid ethane for vitrification.

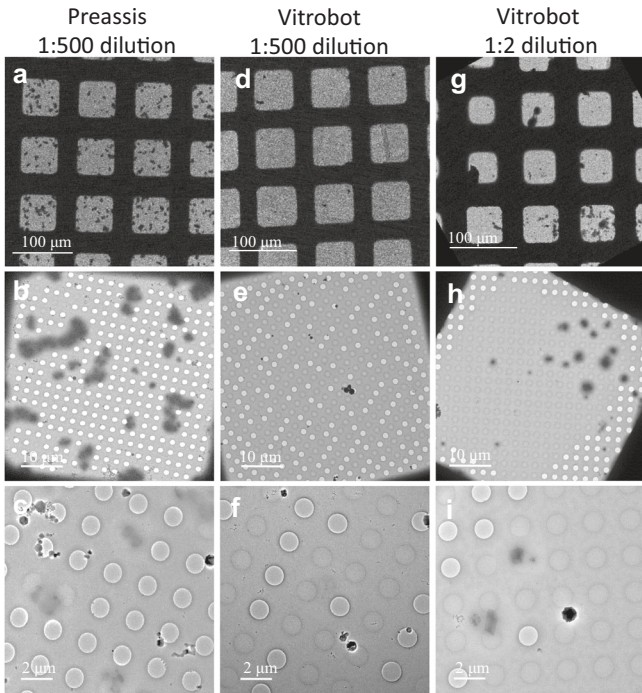

**Fig. 2 Comparison of the density of tetragonal lysozyme crystals on EM grids prepared by Preassis and Vitrobot. a–c** Typical TEM images taken from the grid prepared by Preassis using 500× diluted sample. **d–f** Typical TEM images taken from the grid prepared by Vitrobot using 500× diluted sample. **g–i** Typical TEM images taken from the grid prepared by Vitrobot using 2× diluted sample. Each specimen preparation was repeated three times and similar results were obtained. More details of these copies are shown in Supplementary Figs. 4 and 5.

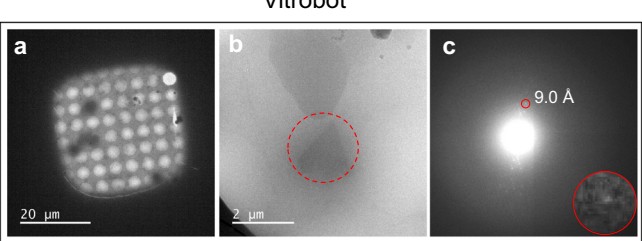

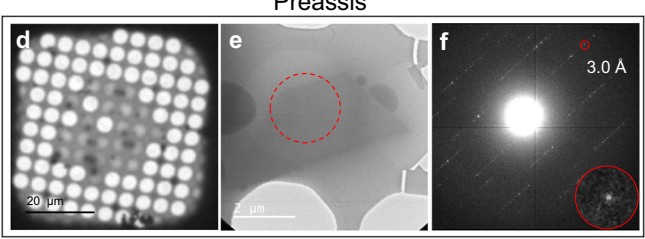

**Fig. 3 Comparison of ice thickness and data resolution obtained from the specimens prepared by Vitrobot and Preassis. a–f** TEM images (**a–b**, **d–e**) and ED patterns (**c**, **f**) taken from specimens prepared from the same viscous R2lox crystal suspension by Vitrobot and Preassis, respectively. The ice thickness can be reduced and ED data resolution was improved from 9.0 Å (**c**) to 3.0 Å (**f**) by applying Preassis. ED patterns (**c**, **f**) were taken from the areas marked as red circles in images **b** and **e**. The insets in images **c** and **f** show a close up of the spots indicated by the red circles. This comparison was based on the results from at least ten repeats using Vitrobot and three repeats using Preassis.

crystals are also removed together with the liquid. Because the number of crystals in a crystallization drop can be very low in many cases, such a unique advantage of Preassis can be significant. Furthermore, sufficient crystal density on the EM grid will benefit data collection by serial electron crystallography, where thousands of crystals are needed to achieve high data completeness[22].

In order to compare the data quality obtained on EM grids prepared by Vitrobot and Preassis, MicroED data were collected from more than ten crystals on each grid (shown in Supplementary Fig. 5). The three best MicroED datasets from each grid were selected for comparison, and the data statistics are given in Supplementary Tables 1 and 2, respectively. The data resolution (with I/sigma ≥ 1), I/sigma, $R_{meas}$, and $CC_{1/2}$ are on average 2.67(11) Å, 5.0(5), 0.314(29), and 0.966(7) from the grids prepared by Vitrobot, and 2.57(8) Å, 5.1(7), 0.320(28), and 0.963(14) from the grids prepared by Preassis. They are very similar. Our results show the data quality is comparable for specimens prepared by both methods when suitable ice thickness and crystal density are achieved.

Another important advantage of Preassis is its ability to handle protein crystals grown in viscous buffers. We performed a systematic comparison of the ice thickness of the grids prepared by Vitrobot and Preassis, and studied the influence of humidity on ice thickness (Supplementary Fig. 6). A suspension of microcrystals of an inorganic zeolite ZSM-5 mixed with 40% PEG 400 was used for this study. The ice thickness was compared based on the transparency of the grids as described in Supplementary Fig. 7. We found humidity had a large impact on the ice thickness for grids prepared by Vitrobot. At ambient humidity (35%), a majority of grid squares are transparent. At high humidity (80

and 100%), the number is reduced by nearly four times and very few grid squares are useful (Supplementary Fig. 6a–c), which makes it difficult to find regions with suitable ice thickness. For the grids prepared by Preassis, nearly all grid squares are transparent, and no significant influence of humidity was found (Supplementary Fig. 6d, e). At both 35 and 80% humidity, grid squares with suitable ice thickness could be found throughout almost the entire grids prepared by Preassis. This could be because the increased humidity decreases the water absorption ability of the filter paper. With Preassis, in such a case, the pressure can assist the liquid removal and therefore the humidity has less influence on Preassis than that on Vitrobot. Our results show that Preassis is more efficient in removing viscous liquid and less affected by high humidity compared to Vitrobot.

Practically, a protein sample can have both low crystal density and high viscosity, making the specimen preparation extremely difficult by Vitrobot. One such example was crystals of *Sulfolobus acidocaldarius* R2-like ligand-binding oxidase (R2lox) grown with 44% PEG 400[3]. It was difficult to remove sufficient amount of liquid by Vitrobot with extreme blotting conditions (2 layers of filter paper on each side, strong blotting force (16), and long blotting time (10 s)), as shown in Fig. 3a–c. Only a few grid squares were electron beam transparent, but the ice layers were too thick and very few crystals could be found, making it extremely difficult to obtain sufficient MicroED data with good diffraction quality. The diffraction quality of MicroED data of R2lox couldn't be improved and the R2lox project got stuck for more than 1 year until the Preassis method was applied. Using Preassis with 30.7 mbar pressure and Quantifoil grid R 3.5/1, the viscous liquid was efficiently removed obtaining a lot of grid squares suitable for searching crystals for MicroED data collection, as shown in Fig. 3d–f. Owing to the reduced vitrified ice thickness and increased crystal density on the grid, the resolution of ED data was significantly improved from 9.0 to 3.0 Å (as shown by ED patterns in Fig. 3c, f), and sufficient and high-quality

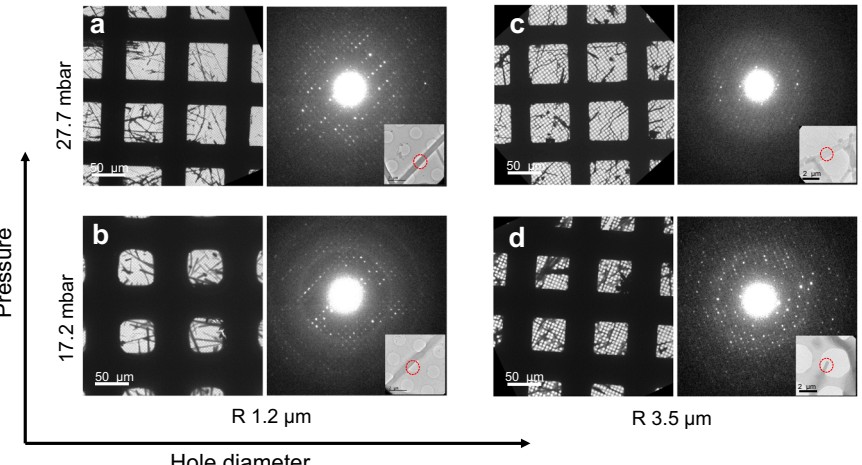

**Fig. 4 Adjustment of vitrified ice thickness by tuning pressure and choosing EM grids with different carbon hole sizes. a**-**d** Representative low magnification TEM images and typical ED patterns taken from orthorhombic lysozyme crystal specimens prepared under different pressures (17.2 and 27.7 mbar) and Quantifoil grids with different hole sizes (R 1.2/1.3 and R 3.5/1). Insets display the areas of lysozyme crystals (marked by red circles) from which the diffraction patterns were taken. This comparison shows both the pressure and hole size of the grid play important roles in ice thickness adjustment. Each specimen preparation condition was repeated three to five times and the results are similar.

MicroED data were collected from one single TEM grid prepared by Preassis. A typical MicroED dataset of R2lox is shown in Supplementary Movie 1. The successful preparation of thin vitrified cryo-EM grids by Preassis made it possible to determine the structure of R2lox, the first novel protein structure solved by MicroED[3].

It is worth noting that, unlike single-particle cryo-EM experiments where 10,000 to 100,000 of particles are required for a complete structure determination, MicroED experiments require only a few good crystals. In some cases, a single microcrystal is sufficient for structure determination[23]. Therefore, successful MicroED specimen preparation is to ensure there are sufficient microcrystals covered by thin ice on several grid squares so that high-quality MicroED data can be collected. Many parameters can affect the ice thickness on the grid, including the carbon hole size of EM grids, pressure and time applied, and the types of filter papers. In this work, the influences of pressure and hole sizes were investigated by combining TEM images and ED patterns. Each specimen preparation condition was repeated 3–5 times. Fig. 4 illustrates how the pressure and hole size affect the overall ice thickness on the EM grid and the resulting quality of ED patterns using orthorhombic lysozyme crystals as an example[24]. By applying a low pressure of 17.2 mbar in a combination of EM grids with a small hole size of 1.2 μm (R 1.2/1.3 Quantifoil grids), usable specimens could be obtained. However, the vitrified ice was relatively thick as seen by the reduced transparent area in each grid square and strong ice rings in corresponding ED patterns (Fig. 4b). A better EM grid with thinner ice layers could be obtained by slightly increasing the pressure to 27.7 mbar, where the sharp edges of grid squares are visible in the TEM images (Fig. 4a). With thinner ice, it was easier to find suitable crystals for MicroED data collection, and ice rings were eliminated in the ED patterns. Consequently, the quality of MicroED data was improved in terms of resolution and signal-to-noise ratio.

Furthermore, the carbon hole size of the EM grid has a strong influence on the ice thickness. Instead of increasing the pressure, thinner ice layers can also be obtained by using EM grids with larger holes because it is easier for liquids to pass through. When a R 3.5/1 grid with a hole diameter of 3.5 μm was used (Fig. 4d), the ice became thinner compared to that of R 1.2/1.3 grids prepared under the same pressure (17.2 mbar, Fig. 4b). High resolution reflections could be obtained in the corresponding ED

pattern and no ice rings are observed. On the other hand, a combination of high pressure (27.7 mbar) with a large hole size (R 3.5/1 grid) led to slight dehydration of the lysozyme crystals, as indicated by reduced resolution in the ED pattern (Fig. 4c). Our results show that optimal ice layer thicknesses can be obtained under several conditions; both combinations with high pressure/small hole size (Fig. 4a) and low pressure/large hole size (Fig. 4d) produced optimal EM grids for MicroED data collection. This is significant because it allows us to select EM grids according to the size and shape of the crystals, and fine-tune the ice thickness by adjusting the pressure. Ideally, the carbon hole size of the grid should be slightly smaller than or comparable to the largest crystal dimensions to maximize the hole areas and at the same time prevent the loss of crystals through the holes. This is particularly important when the initial density of crystals is low. We anticipate that the size and shape of microcrystals may also affect the ice thickness. Therefore, compared to the optimal pressure (27.2 mbar) for the preparation of orthorhombic lysozyme crystals with rod-like morphology on the R 1.2/1.3 grids, a higher pressure (37.2 mbar) was needed to achieve similar ice thickness and MicroED data quality for the tetragonal lysozyme crystals (Fig. 2a–c). It is observed that despite the absence of vitrified ice in the neighbouring empty holes, most crystals are still embedded in ice, showing microcrystals attract more liquids than the surface of the carbon film do presumably caused by the stronger surface tension. Consequently, the applied pressure needs to be adjusted according to the size and shape of the crystals.

For non-viscous samples (e.g., no high molecular weight polymers in the buffer), usable specimens can be prepared by Preassis even without applying pressure if other parameters (e.g., hole size and time) are carefully chosen (Supplementary Fig. 8). If the crystal suspension is viscous due to e.g. high concentration or high molecular weight of PEGs in the buffer, high pressures (e.g., >180 mbar) are required. To systematically study the specimen preparation under different viscosities, several experiments were conducted with various concentrations of PEG 6000 mixed with microcrystals. Submicron-sized crystals of an inorganic sample zeolite ZSM-5 were used instead since it is difficult to obtain the same type of protein crystals from mother liquids of different viscosities. The results are shown in Supplementary Fig. 9. We found that, for crystal suspension with PEG 6000 lower than 25%, usable grids could be obtained without applying

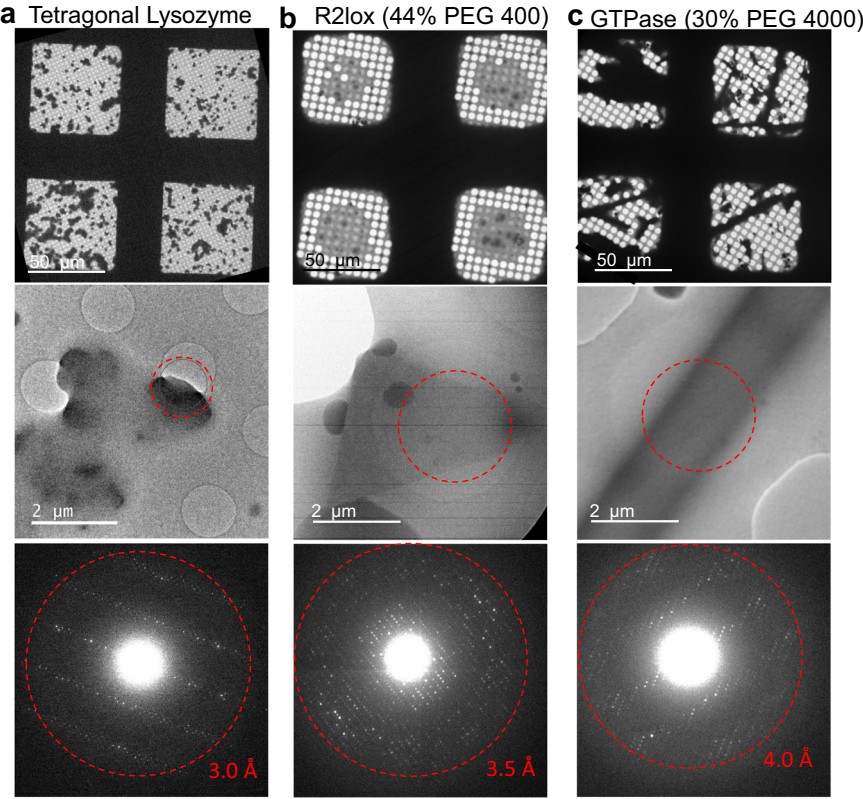

**Fig. 5 Examples of specimens prepared successfully by Preassis from non-viscous to viscous buffer conditions. a** MicroED specimens prepared from tetragonal lysozyme crystal sample under non-viscous condition (without high molecular weight polymer in the buffer) using grid R 1.2/1.3 and pressure of 37.2 mbar. **b, c** MicroED specimens prepared from R2lox and GTPase crystal samples under viscous condition with 44% PEG 400 and 30% PEG 4000, respectively. The pressure used for R2lox specimen is 30.7 mbar, and GTPase is 181 mbar. Grids R 3.5/1 were used for both experiments. ED patterns were taken from the crystal areas highlighted by red circles. Each specimen preparation was repeated at least three times and the results are similar.

pressure when grids with large holes were used (e.g., Quantifoil R 3.5/1, Supplementary Fig. 9f, g). However, the amorphous ice on grids prepared under those conditions was relatively thick in most grid squares, which reduces the contrast of the crystals and makes it difficult to find suitable crystals for MicroED data collection. By applying 181 mbar pressure, the specimens could be improved (Supplementary Fig. 9b, c). When the crystal suspension was very viscous, such as containing 35% PEG 6000, high pressure was necessary in order to obtain usable grids (Supplementary Fig. 9d, h). In reality, the pressure could be further increased to reduce the vitrified ice thickness for the cases as shown in Supplementary Fig. 9c, d. In conclusion, grids with large hole sizes and high pressure are recommended for viscous crystal suspensions. In cases of crystals grown in extremely viscous buffers such as LCPs, it is difficult to obtain usable EM grids by Preassis alone, a combination with other methods may be necessary, e.g., adding detergents, oils, or lipase[25] to decrease the viscosity of the LCP.

Even if a good or usable specimen is obtained, finding ideal grid squares and good crystals for data collection is also important. This is because the thickness distribution of the amorphous ice layer is often not homogeneous across the grid, sometimes is not uniform within the same grid square. For crystals grown in non-viscous mother liquors, ideal grid squares and crystals have the following features: (1) grid squares with clear and sharp edges when viewed in low magnification (Fig. 5a), indicating the vitrified ice is very thin, (2) crystals found in these squares with blurred edges on the carbon film, indicating the crystals are still hydrated, and (3) crystals hanging over the empty holes with relatively clear edges, indicating the ice is very thin. It is desirable to collect MicroED data from these types of crystals, and at the

regions where they are hanging over the holes. In these cases, carbon background is avoided, the ice thickness is minimized, and the crystals are usually still hydrated. A representative example including images and diffraction patterns is shown in Fig. 5a. When the buffer is viscous (e.g., 44% PEG 400 and 30% PEG 4000), the grid squares no longer show sharp edges and transparent areas are reduced compared to those prepared under non-viscous conditions, as shown in the low-magnification images in Fig. 5b, c. Under such circumstances, the ideal crystals for MicroED data collection are still those hanging over the holes (crystal images and ED patterns in Fig. 5b, c) despite the surrounding vitrified ice. Furthermore, for both viscous and non-viscous cases, most holes in the optimized grid squares are empty without vitrified ice except for the holes with crystals nearby or on top of the holes.

In conclusion, Preassis is a simple and promising method for preparing MicroED specimens. Our results demonstrate that the ice thickness can be adjusted by tuning pressure and selecting grids with different hole sizes. More importantly, Preassis allows the preparation of MicroED specimens for crystals grown in highly viscous media and keeps a relatively high density of crystals on the EM grid. Here we provide a guideline on how to select parameters for specimen preparation of a new protein crystal sample using Preassis. Common features of good grid squares and crystals for high-quality MicroED data collection are discussed, which will be important for future automation of data collection and high-throughput structure determination by MicroED. Preassis is simple and can be easily implemented in all cryo-EM labs. New implementations of Preassis, such as developing automation and adding an environmental chamber, may

improve the throughput and reproducibility of MicroED specimen preparation. It is important to mention that MicroED experiments require small microcrystals, preferably within ~500 nm in at least one of the crystal dimensions. While mechanical crystal segmentation[10] or cryo-FIB milling[11–14] can be applied to reduce the size of crystals too large for MicroED, these methods are not optimal. More research is needed to find optimal conditions to directly grow small microcrystals. Furthermore, it is also important to develop new screening methods for such microcrystals because they are hardly visible under light microscopes. We believe, by overcoming the bottleneck for MicroED specimen preparation, more protein structures can be studied by MicroED. Our Preassis method will significantly widen the application of MicroED.

## Methods

**Protein crystallization**. Hen egg-white orthorhombic lysozyme microcrystal sample was produced as described by Xu et al.[24]. In brief, the protein was crystallized using the hanging-drop vapour-diffusion method where 2 µl of an 8 mg/ml protein solution in 10 mM Tris-HCl pH 8.0 was mixed with 2 µl of reservoir solution consisting of 1 M potassium nitrate, 0.1 M sodium acetate trihydrate pH 3.4, to produce thin fibrous needle-shaped crystal clusters within 48 h at 21 °C.

Hen egg-white tetragonal lysozyme microcrystals were grown by a modified batch method as described previously[26,27]. In brief, Hen egg-white lysozyme (Calzyme laboratories inc.; 50,000 U/mg) was diluted by 0.1 M potassium phosphate buffer pH 6.24–8 mg/ml. The protein solution was then mixed with a reservoir solution (0.5 M acetate/acetic acid,15% (w/v) NaCl, 6% PEG 6000, pH 4.0) by a volume ratio of 1:3 to obtain batches of 1 ml using 24-well Hampton Research VDX plates. Tetragonal lysozyme crystals were obtained after 24 h at 21 °C and then stored in 8% (w/v) NaCl, 0.1 M sodium acetate buffer, pH 4.0. Submicron-sized crystals suitable for MicroED experiments were obtained by fragmentation using glass beads (0.5 mm). The original crystal suspension was further diluted by 500 times using the same buffer and used for Preassis and Vitrobot specimen preparations. A crystal suspension diluted by only two times from the original one was also used for Vitrobot specimen preparation, in order to keep enough crystals on the EM grids.

Crystal sample of the R2lox protein (*Sulfolobus acidocaldarius* R2-like ligand-binding oxidase) was produced as described by Xu et al.[3]. In brief, the protein was crystallized using the hanging-drop vapour-diffusion method where a volume of 2 µl of an 8 mg/ml protein solution was mixed with 2 µl of reservoir solution consisting of 44% (v/v) PEG 400, 0.2 M lithium sulfate and 0.1 M sodium acetate pH 3.4. Plate-like crystals grew within 48 h at 21 °C.

Crystallization of human dynamin I BSE-GTPase was performed using sitting-drop vapour-diffusion at 20 °C. Screening of crystallization conditions was carried out with JBScreen Classic 1 and 2 (Jena Bioscience, Jena, Germany). Two microlitres of 9 mg/ml protein solution in 20 mM HEPES-NaOH pH 7.5 and 150 mM NaCl buffer were mixed with an equal volume of reservoir solution and equilibrated against 0.5 ml of reservoir solution. The reservoir solution contained 30% (w/v) PEG 4000, making the solution highly viscous. The initial tiny needle-like crystals grew to micrometre size within a day and are suitable for MicroED experiments.

**Specimen preparation by Preassis**. For MicroED specimen preparation of orthorhombic and tetragonal lysozyme crystals, quantifoil grids R 1.2/1.3 and R 3.5/1 were used. The grids were glow discharged with 20 mA current for 60 s by PELCO easiGlow™ 9100. A droplet of 3 µl (the same for all other specimen preparations in this work) of crystal suspensions was applied to a glow-discharged grid. The pressures used were 17.2 mbar and 27.7 mbar for the orthorhombic lysozyme sample, 0 mbar and 37.2 mbar for the tetragonal lysozyme sample. The time from applying the sample on the grid to manually plunge freezing was ~5 s for both specimen preparations. All specimen preparations were carried out at room temperature (~20 °C) with ambient humidity (~35%) (see Figs. 4, 5a, and Supplementary Figs. 3, 4a–c, 8) or 80% humidity (see Fig. 2a-c and Supplementary Fig. 5a–c).

For R2lox and GTPase crystal samples, quantifoil grids R 3.5/1 were used. The glow-discharge conditions were the same as above. The pressure and time used for R2lox and GTPase specimens were 30.7 mbar, 5 s and 181 mbar, 10 s, respectively.

Since it is difficult to get the same type of protein crystals grown in mother liquors of different viscosities, micro-sized inorganic crystals (ZSM-5[28]) mixed with different concentrations (15%, 25% and 35% (w/v)) of PEG 6000 and 40% (v/v) PEG 400 were used. Quantifoil grids R 1.2/1.3, R 2/1 and R 3.5/1 were used in these experiments and were glow discharged as above. For the specimens with PEG 6000, the pressure was either 0 mbar or 181 mbar, the time from applying the sample on the grid to manually plunge freezing was ca 10 s and the temperature and humidity were ~20 °C and ~35%, respectively. For the specimens with 40% PEG 400, the pressure was 180 mbar and the time was ca 5 s. All specimen preparations were

carried out at room temperature (~20 °C) with ambient humidity (~35%, see Supplementary Fig. 6d) or 80% humidity (see Supplementary Fig. 6e).

**Specimen preparation by Vitrobot Mark IV (Thermo Fisher Scientific)**. For MicroED R2lox specimens, a droplet of 3 µl of R2lox crystal suspension was applied to a glow-discharged (60 s, 20 mA, PELCO easiGlow) Quantifoil grid R 3.5/1. The operation parameters of the Vitrobot Mark IV were 4 °C, 100% humidity, 10 s blotting time, two layers of blotting papers on each pad, and blotting force 16.

For MicroED tetragonal lysozyme specimens, a droplet of 3 µl of lysozyme crystal suspension was applied to a glow-discharged (60 s, 20 mA, PELCO easiGlow) Quantifoil grid (R 1.2/1.3). The operation parameters of the Vitrobot Mark IV (Thermo Fisher Scientific) were 20 °C, 80% humidity, single blot, 5 s blotting time, 1 blotting paper on each pad, and blotting force 16.

For ZSM-5 crystal suspension containing 40% (v/v) PEG 400, quantifoil grids R 3.5/1 were used and glow-discharged as above. A 3 µl droplet of the crystal suspension was applied to a glow-discharged grid. The operation parameters of the Vitrobot were single blot, 5 s blotting time, 1 blotting paper on each pad, and blotting force 5. All specimen preparations were conducted at room temperature (20 °C), with either ambient (35%), 80% or 100% humidity, as shown in Supplementary Fig. 6a–c.

**TEM imaging and ED data collection**. TEM images (except for those in Fig. 2 and Supplementary Fig. 5) and ED patterns were collected under the cryogenic condition on a JEOL JEM-2100LaB$_6$ TEM (200 kV) using a Gatan 914 cryo-transfer holder. All the TEM images were taken at the image mode on a Gatan Orius camera (2048 × 2048). An ultra-low magnification (×50) was used to image the entire grid. To image grid squares, a magnification range of ×100–300 and a pixel size range of 0.15–0.9 µm/pixel were used. To image crystals within a grid square, a magnification range of ×12,000–2500 and a pixel size range of 6.6–32 nm/pixel were used. DigitalMicrograph was used for the image analysis for studying the vitrified ice thickness of specimens. Selected area ED patterns and MicroED data of R2lox (Supplementary Movie 1) were recorded by a fast Timepix hybrid pixel detector (512 × 512, Amsterdam Scientific Instruments). The conditions used for the data collections were as follows: spot size 3, camera length 80 cm/100 cm, and exposure time 1 s/2 s per frame. MicroED data of R2lox were collected by continuously rotating the crystal whilst ED frames were simultaneously recorded. The rotation speed of the goniometer was 0.45°s$^{-1}$. The dose rate was estimated to be 0.10 e$^-$Å$^{-2}$s$^{-1}$. The software used for data collection was *Instamatic*[29].

TEM images in Fig. 2 and Supplementary Fig. 5 and MicroED data in Supplementary Tables 1 and 2 were collected on a Themis Z microscope (300 kV) equipped with a monochromator and a Gatan OneView camera (4096 × 4096). A Gatan Elsa™ 698 cryo-transfer holder was used to keep the grids at cryogenic condition. Different magnifications were used to image the entire grid (100×, 339.0 nm/pixel, 1024 × 1024 with 4× binning) and crystals within a grid square (660×, 13.0 nm/pixel, 4096 × 4096). Atlas images (left column of Supplementary Fig. 5) were obtained by stitching 36 images (magnification ×100) using *Instamatic*[29]. MicroED data were collected by *InsteaDMatic*[30] on the Gatan OneView camera using the in situ data capture mode (1024 × 1024, 4× binning). The parameters used for MicroED data collection were as follows: spot size 11, Mono −50, camera length 2.3 m, dose rate 0.03 e$^-$Å$^{-2}$ s$^{-1}$, rotation speed 0.57°s$^{-1}$, and exposure time 2 s/frame. The data were then processed and analyzed by XDS[31].

**Reporting summary**. Further information on research design is available in the Nature Research Reporting Summary linked to this article.

## Data availability

A raw MicroED data of R2lox, collected from the specimen prepared by Preassis, is attached as Supplementary Movie 1. Raw tetragonal lysozyme MicroED data (Supplementary Table 1 and Table 2) are available from the SBGrid Data Bank (doi: 10.15785/SBGRID/842). Other data are available from the corresponding authors upon reasonable request. Source data are provided with this paper.

## Code availability

There are no new codes/scripts included in this manuscript. All software used in this manuscript are previously published and the corresponding references are cited.

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

## Acknowledgements

The project is supported by the Knut and Alice Wallenberg Foundation (2012.0112 and 2018.0237, X.Z.; 2017.0275, M.H.), the Swedish Research Council (2017-04018, M.H.; 2017-05333, H.X.; 2019-00815, X.Z.) and the Science for Life Laboratory through the pilot project grant Electron Nanocrystallography, and MicroED@SciLifeLab. The authors acknowledge the Cryo-EM Swedish National Facility jointly funded by the Knut and Alice Wallenberg, Family Erling Persson and Kempe Foundations, SciLife-Lab, Stockholm University, and Umeå University. The authors also thank I. Schlichting at Max-Plank Institute for Medical Research for providing the tetragonal lysozyme sample.

## Author contributions

J.Z. contributed to method development, design and tests of the Preassis setup, specimen preparation, TEM and MicroED data collection, data processing and analysis, initial manuscript preparation and figure preparation. H.X. contributed to project design, method development, specimen preparation, MicroED data collection, and manuscript preparation. M.C. contributed to the project discussion. H.L., H.T. and M.H. prepared protein microcrystals and contributed to the project discussion. X.Z. contributed to project design, the conception of Preassis, and manuscript preparation. All authors contributed to the method discussions and manuscript revisions. H.X. and X.Z. led the project and finalized the manuscript.

## Funding

## Competing interests

The authors declare no competing interests.
