## [Peer Review File · Nature Communications]

A simple pressure-assisted method for MicroED specimen preparationREVIEWER COMMENTS

Reviewer #1 (Remarks to the Author):

In this work, the authors describe a method for micro-crystal electron diffraction (MicroED) sample preparation that uses pressure assisted blotting. This system should be a very useful tool for preparing MicroED samples, which can be a bottleneck to applying the method for structure determination. That being said, there are some concerns with the manuscript as submitted.

-The description of the improvements due to the new sample preparation method are all very qualitative. The thickness is said to be thinner, without much other description. While measuring the exact thickness, other descriptions should be added (e.g. how much area was deemed too thick, how many different grids were prepared for each assessment). Also, single diffraction patterns are shown, and diffraction is said to be better with one method over another, however, this could be described quantitatively. Multiple grids should be made by each method, and multiple crystals should be collected from these grids. Then the data could be processed with some metric used for each crystal's resolution limit (e.g. $CC1/2$, I/σ , furthest visible spot). These statistics would give readers a much better idea of how much better the method is for preserving the crystals.

-Page 6 starting at line 101 says "Preassis is crucial for the successful structure determination of R2lox.." This statement is very confusing as this sample seems to have already been determined in reference 3. If this sample was already solved using other methods, then while this method may make things easier, it does not seem to be "crucial". If this method improves the structure quality, then that should be stated and quantified. If this sample preparation method is what was used in reference three (it appears to not have been based on reading reference 3), then this should be mentioned and more discussion on what makes this work novel relative to reference 3.

Page 2 line 29: "attracted large interests..." should be interest

Page2 line 31: "... in the PDB database, all except R2lox.." should be "...in the PDB database and all except R2lox..."

Page 4 line 79: Talking about hole size and saying it can be controlled by the type of EM grid may confuse some readers who may think mesh size (which is the spacing of the EM grid) is what is meant. The authors are referring to hole size of the holey carbon film so it may be useful to use this term instead.

Figure 1 d/e: The crystals look bent in the low mag image. Are these bundles of smaller crystals that give the appearance of bent crystals?

Page 7 line 110: Should mention that in some cases a single crystal is all that is required. Only saying "up to 50" may give the impression that that is common for large numbers to be used when it really depends on the sample.

Page 9 line 155: would be better to say the surface of the carbon attracts the liquid instead of the entire EM grid

Reviewer #2 (Remarks to the Author):

The paper entitled "A simple pressure-assisted method for MicroED specimen preparation" proposes a new approach to making ideal grids for MicroED data collection of hydrated crystal specimens, by filtration of the sample (microcrystals in suspension) through a Quantifoil grid on filter paper, with gentle suction via a Büchner funnel. This method promises to give the researcher more control on making grids with microcrystals in thin ice, including crystal samples in viscous solutions such as high concentration/molecular-weight PEG, and potentially crystals grown in LCP. This paper represents notable progress in the preparation of samples for MicroED, and therefore is of general interest to researchers in macromolecular crystallography and structural biology.

To be addressed in the revision:

1) Supplementary fig 2 shows a humidity comparison between Preassis vs. Vitrobot, however, the result for Preassis at 100% humidity is not shown. Such a setup should not be difficult to emulate for Preassis; it would be interesting to see how Preassis at 100% humidity compares with the Vitrobot at 100% humidity. This particular condition would enable imaging/MicroED of certain crystals that are more stable in higher humidity.

2) The protocol mentions a "Munktell #110067 or similar" but I cannot find specifications online of this particular type of filter paper, except that is of "grade 3". There are different grades/types of filter paper that one can use with varying results. The type of filter paper is critical, for example, as one with a coarse/loose fiber grain may puncture the grid upon aspiration. Therefore more information on the ideal filter paper specifications would be beneficial in this regard.

3) Microscope parameters such as magnification/pixel size are not present in the section "TEM image collection", though they (diffraction parameters) are mentioned for the MicroED experiments in the Methods section. Because figs 1-3 and supplementary figs 2-6 have scale bars in the images, then the magnification/pixel size parameters are known, and should ideally be included in the Methods section.

4) It appears that the very low-mag images (in supplementary figs 3-6) of the entire TEM grid were taken in diffraction mode with the diffraction focus set to spread the beam from the focused (condensed) beam, hence the pincushion distortion effect. In this case, the scale bar is highly inaccurate across the image due to extreme distortion, and is better left out. For the intermediate low-mag images (e.g., supplementary fig 3d-f), where the distortion is minimal, the scale bar is applicable because it is generally accurate. Either the cause of the distortion should be mentioned in the text, or preferably, these images should be taken in an imaging lens mode (i.e., "LOW MAG" mode on the JEOL), where the pixel size is known and where there is minimal distortion in the image to render an accurate scale bar.

Reviewer #3 (Remarks to the Author):

The manuscript by Zhao and coworkers describes Preassis, a method to prepare crystalline samples for micro-crystal electron diffraction (MicroED) measurements. Sample preparation has long been one of the most finicky aspects of MicroED as evidenced by several publications on the subject, many of which are cited in the manuscript. The difficulty lies in removing just the right amount of buffer around the sample before freezing it and doing so without damaging the crystal lattice. If the ice around the crystals is too thin, the sample may run dry during subsequent data collection; if it is too thick, the beam may not be able to penetrate the sample or the weak high resolution reflections may disappear in the noise contribution of the solvent. If the lattice is damaged, the diffractive power of the crystal will be diminished.

The strength of Preassis lies in its simplicity: it can be implemented using readily available laboratory components. The method is mostly controlled by the user's choice of grid, which determines the size and distribution of holes, and the pressure applied to the backside of the grid. The idea is to tune these two variables such that excess solvent is removed once the sample is dispensed on the grid but crystals remain intact. Based on their success with crystals of two forms of lysozyme, GTPase, ZSM-5, and R2lox crystallized in a viscous buffer, the authors conclude that Preassis is applicable to a wide range of protein crystals and buffer conditions. The case of R2lox is particularly compelling, since it had previously resisted successful structure solution due to a thick, impenetrable solvent layer that could not be adequately blotted owing to its viscosity. The thick solvent would extinguish the weak signal from the faint spots during data collection, leading to a marked decrease in resolution.

The manuscript is well-written and clearly describes the method to the reader. The work on Preassis is original, and the method appears to offer clear advantages to the commonly used blotting and

plunging techniques of sample preparation. In the cases shown, Preassis handles viscous solvent more gracefully and was found to preserve tenfold more crystals on the prepared grids than the ubiquitous Vitrobot and it can be implemented at the fraction of the price.

Major comments

Owing to the strong interaction between the electrons and the sample, MicroED excels at obtaining useful diffraction data from minuscule crystals but struggles when crystals are too big. Since finding crystallization conditions that yield smaller crystals is often unfeasible, this is probably the biggest problem in sample preparation for MicroED. The literature contains methods to address this problem by fragmenting crystals or using a focused ion beam to mill crystals into thin sheets. Neither of these techniques are optimal, as one can be quite blunt and the other is often very time-consuming. Sadly, it is also an issue that the manuscript does not address at all, and this dampens the enthusiasm for the method and makes statements about the universality of the method appear somewhat inflated. The authors may assume that the large-crystal problem is solved either pre- or post-Preassis, in which case a brief discussion of this problem should be included in the text.

Other comments

The manuscript rightly points out that no two samples behave identically, and that each new sample will require parameters individually tuned to the size and shape of the crystals and the viscosity of the solvent around it. For conventional sample preparation with a plunger, this optimization can be quite costly in terms of both time and sample consumption. Even though the lysozyme example makes the relationship between the parameter values and the outcome of Preassis appear very straightforward, the ensuing two-dimensional search space is still substantial. Some information on how much time one may expect to spend on optimizing sample preparation parameters would not be amiss.

Further information about the difficulty of parameter tuning is particularly relevant to the claims of future automation, where this kind of optimization would have to be carried out without human intervention. None of the sample preparation methods currently described in the literature are particularly amenable to automation, and it is not immediately obvious what form such an implementation could take; a more detailed speculation on this topic would be appreciated.

The amount of time the sample is left on the filter paper is another parameter in Preassis. Even though it varies by a factor of two for the examples given, it is not discussed much in the text. Presumably, the proposed enhancement of using a humidity chamber is related to this time, so it must have some impact on the result. A related question concerns the speed at which the grid is plunged into liquid ethane. Are there any effects of uneven vitrification, since grids in Preassis are plunged manually?

Minor comments and suggested improvements

Were crystals ever observed to break due to the applied pressure? Presumably, this does happen, as the tetragonal lysozyme crystals shown in the manuscript rarely extend into the grid holes, even when they are sitting close their edges.

Depending on what constitutes a protein versus a peptide, there are arguably more than 16 unique MicroED structures in the PDB as of this writing. Since this number is rapidly changing, it may be better to give a more approximate count in the text.

Response letter to the Reviewers's comments on "A simple pressure-assisted method for MicroED specimen preparation" by Zhao et al.

We would like to give our sincere thanks to the Reviwers for the valuable comments. Please find the point-to-point responses below, which are given in blue. The changes and new text in the revised manuscript are marked in yellow.

Reviewer #1 (Remarks to the Author):

In this work, the authors describe a method for micro-crystal electron diffraction (MicroED) sample preparation that uses pressure assisted blotting. This system should be a very useful tool for preparing MicroED samples, which can be a bottleneck to applying the method for structure determination. That being said, there are some concerns with the manuscript as submitted.

-The description of the improvements due to the new sample preparation method are all very qualitative. The thickness is said to be thinner, without much other description. While measuring the exact thickness, other descriptions should be added (e.g. how much area was deemed too thick, how many different grids were prepared for each assessment). Also, single diffraction patterns are shown, and diffraction is said to be better with one method over another, however, this could be described quantitatively. Multiple grids should be made by each method, and multiple crystals should be collected from these grids. Then the data could be processed with some metric used for each crystal's resolution limit (e.g. $CC_{1/2}$, I/σ , furthest visible spot). These statistics would give readers a much better idea of how much better the method is for preserving the crystals.

Response: We thank the reviewer for the valuable suggestions. We have now performed additional experiments to make more quantitative and systematic studies on the ice thickness of the grids prepared by Preassis and Vitrobot, and the influence of humidity on ice thickness. A systematic comparison of the MciroED data quality obtained from the specimens prepared by these two methods, in terms of data resolution, I/σ , R_{meas} , and $CC_{1/2}$, has now been added in the revised manuscript.

1) A comparison of the ice thickness of the grids prepared by Preassis and Vitrobot has been done by using the same crystal suspension with 40% PEG 400 at different humidity (35% and 80%). Each specimen preparation condition has been repeated at least four times and new figures have been added in the Supplementary information (Supplementary Figs. 6 and 7). The ice thickness was described quantitatively by the total pixel number of the transparent area of each grid as shown in Supplementary Fig. 7. We have added a new paragraph in the main text on page 7 to address this issue, as given below:

'Another important advantage of Preassis is its ability to handle protein crystals grown in viscous buffers. We performed a systematic comparison of the ice thickness of the grids prepared by Vitrobot and Preassis, and studied the influence of humidity on ice thickness (Supplementary Fig. 6). A suspension of microcrystals of an inorganic zeolite ZSM-5 mixed with 40% PEG 400 was used for this study. The ice thickness was compared based on the transparency of the grids as described in Supplementary Fig. 7. We found humidity had a large impact on the ice thickness for grids prepared by Vitrobot. At ambient humidity (35%), a

majority of grid squares are transparent. At high humidity (80% and 100%), the number is reduced by nearly 4 times and very few grid squares are useful (Supplementary Figs. 6a-c), which makes it difficult to find regions with suitable ice thickness. For the grids prepared by Preassis, nearly all grid squares are transparent, and no significant influence of humidity was found (Supplementary Figs. 6d-e). At both 35% and 80% humidity, grid squares with suitable ice thickness could be found throughout almost the entire grids prepared by Preassis. This could be because the increased humidity decreases the water absorption ability of the filter paper. With Preassis, in such a case, the pressure can assist the liquid removal and therefore the humidity has less influence on Preassis than that on Vitrobot. Our results show that Preassis is more efficient in removing viscous liquid and less affected by high humidity compared to Vitrobot.’

2) Additional experiments were performed to quantitatively study the quality of MicroED data collected from the specimens prepared by Vitrobot and Preassis. These experiments were performed using tetragonal lysozyme crystal suspensions, instead of a viscous protein crystal suspension. This is because that the crystal density of the viscous crystal suspension was too low for Vitrobot, and the grids prepared by Vitrobot were far from ideal. MicroED data sets were collected from multiple crystals on several grids prepared by Vitrobot and Preassis. Data statistics have been added in Supplementary Table 1 and Table 2. We have added a new paragraph in the main text on page 6 to address this comparison, as given below:

‘In order to compare the data quality obtained on EM grids prepared by Vitrobot and Preassis MicroED data were collected from more than 10 crystals on each grid (shown in **Supplementary Fig. 5**). The three best MicroED datasets from each grid were selected for comparison, and the data statistics are given in **Supplementary Tables 1 and 2**, respectively. The data resolution (with $I/\sigma \geq 1$), I/σ , R_{meas} , and $CC_{1/2}$ are on average 2.67(11) Å, 5.0(5), 0.314(29), and 0.966(7) from the grids prepared by Vitrobot, and 2.57(8) Å, 5.1(7), 0.320(28), and 0.963(14) from the grids prepared by Preassis. They are very similar. Our results show the data quality is comparable for specimens prepared by both methods when suitable ice thickness and crystal density are achieved.’

-Page 6 starting at line 101 says “Preassis is crucial for the successful structure determination of R2lox.” This statement is very confusing as this sample seems to have already been determined in reference 3. If this sample was already solved using other methods, then while this method may make things easier, it does not seem to be “crucial”. If this method improves the structure quality, then that should be stated and quantified. If this sample preparation method is what was used in reference three (it appears to not have been based on reading reference 3), then this should be mentioned and more discussion on what makes this work novel relative to reference 3.

Response: We apologize for the confusion of the statement. In fact, the idea of using pressure-assisted back-side blotting to remove the excess liquid came when we could not obtain good EM grids of R2lox microcrystals (grown with 44% PEG 400) by Vitrobot. In reference 3, our main focus was to solve the first novel protein structure of R2lox by MicroED and we only mentioned “**The excessive liquid was removed by manual back-side blotting**” and did not disclose the Preassis method. The entire sample preparation section in reference 3 is shown below:

‘Sample preparation

A cryo-EM sample of SaR2lox was prepared by freezing the crystals in a thin layer of vitrified ice. A thin and uniform vitrified ice layer is crucial for obtaining MicroED of a high signal-to-noise ratio. Meanwhile, the ice layer has to protect the crystals from being dehydrated under a high vacuum inside a TEM. The 4- μ l hanging drop was deposited onto a QUANTIFOIL R 3.5/1 (300 mesh) Cu holey carbon TEM grid. The excessive liquid was removed by manual back-side blotting. The grid was then rapidly plunge-frozen in liquid ethane. We note that the automated blotting and vitrification routine using a FEI Vitrobot Mark IV was not efficient in removing the viscous mother liquid while leaving a sufficient number of crystals on the TEM grid.’

The current manuscript focuses on the Preassis method, in which systematic studies have been carried out to make a more quantitative comparison of the EM grids prepared by Preassis and Vitrobot. The manuscript also provides a detailed protocol for setting up and using Preassis. We have modified the description in the main text accordingly to make the writing more clear. The sentence ‘Preassis is crucial for the successful structure determination of R2lox, the first novel protein structure solved by MicroED³’ has been changed to ‘The successful preparation of thin vitrified cryo-EM grids by Preassis made it possible to determine the structure of R2lox, the first novel protein structure solved by MicroED³’.

Page 2 line 29: “attracted large interests...” should be interest

Response: We have made the corresponding change.

Page2 line 31: “... in the PDB database, all except R2lox.” should be “...in the PDB database and all except R2lox...”

Response: We have made the corresponding change.

Page 4 line 79: Talking about hole size and saying it can be controlled by the type of EM grid may confuse some readers who may think mesh size (which is the spacing of the EM grid) is what is meant. The authors are referring to hole size of the holey carbon film so it may be useful to use this term instead.

Response: We have replaced “hole size” with “carbon hole size” in most places throughout the manuscript. For example, the sentence ‘While the pressure can be changed continuously, the change of the hole size is done by choosing the type of EM grids.’ has been changed to ‘While the pressure can be changed continuously, the change of the carbon hole sizes is done by choosing the type of holey carbon EM grids.’

Figure 1 d/e: The crystals look bent in the low mag image. Are these bundles of smaller crystals that give the appearance of bent crystals?

Response: The bent crystals were due to the adhesion between the needle-shaped crystal and the carbon film. We have replaced the original images of the needle-shaped lysozyme crystals with images of fragmented tetragonal lysozyme crystals prepared by Preassis and Vitrobot,

respectively. The morphology of the tetragonal lysozyme crystals makes it easier to compare the results. The updated images are shown in Fig. 2 in the main text.

Page 7 line 110: Should mention that in some cases a single crystal is all that is required. Only saying “up to 50” may give the impression that that is common for large numbers to be used when it really depends on the sample.

Response: We thank the reviewer for the comments. The sentence: ‘MicroED experiments require only a limited number of crystals (up to 50)’ has been changed to ‘MicroED experiments require only a few good crystals. In some cases, a single microcrystal is sufficient for structure determination²³’

Page 9 line 155: would be better to say the surface of the carbon attracts the liquid instead of the entire EM grid

Response: We have made the corresponding change.

Reviewer #2 (Remarks to the Author):

The paper entitled "A simple pressure-assisted method for MicroED specimen preparation" proposes a new approach to making ideal grids for MicroED data collection of hydrated crystal specimens, by filtration of the sample (microcrystals in suspension) through a Quantifoil grid on filter paper, with gentle suction via a Büchner funnel. This method promises to give the researcher more control on making grids with microcrystals in thin ice, including crystal samples in viscous solutions such as high concentration/molecular-weight PEG, and potentially crystals grown in LCP. This paper represents notable progress in the preparation of samples for MicroED, and therefore is of general interest to researchers in macromolecular crystallography and structural biology.

To be addressed in the revision:

1) Supplementary fig 2 shows a humidity comparison between Preassis vs. Vitrobot, however, the result for Preassis at 100% humidity is not shown. Such a setup should not be difficult to emulate for Preassis; it would be interesting to see how Preassis at 100% humidity compares with the Vitrobot at 100% humidity. This particular condition would enable imaging/MicroED of certain crystals that are more stable in higher humidity.

Response: We thank the reviewer for the comments and suggestions. We have built a humidity chamber around the Preassis setup which can achieve up to 80% humidity (we were unable to obtain 100% humidity due to the current design of the humidity chamber). A new figure is added in the Supplementary information (see Supplementary Fig. 1). We have performed additional experiments at different humidities using both Preassis and Vitrobot. We made a quantitative comparison on the specimens prepared at ambient humidity (35%) and 80% humidity using both Preassis and Vitrobot. Two new figures are included in the Supplementary information (see Supplementary Figs. 6 and 7). We have added a new paragraph in the main text on page 7 to address this issue, as given below:

‘Another important advantage of Preassis is its ability to handle protein crystals grown in viscous buffers. We performed a systematic comparison of the ice thickness of the grids prepared by Vitrobot and Preassis, and studied the influence of humidity on ice thickness (**Supplementary Fig. 6**). A suspension of microcrystals of an inorganic zeolite ZSM-5 mixed with 40% PEG 400 was used for this study. The ice thickness was compared based on the transparency of the grids as described in **Supplementary Fig. 7**. We found humidity had a large impact on the ice thickness for grids prepared by Vitrobot. At ambient humidity (35%), a majority of grid squares are transparent. At high humidity (80% and 100%), the number is reduced by nearly 4 times and very few grid squares are useful (**Supplementary Figs. 6a-c**), which makes it difficult to find regions with suitable ice thickness. For the grids prepared by Preassis, nearly all grid squares are transparent, and no significant influence of humidity was found (**Supplementary Figs. 6d-e**). At both 35% and 80% humidity, grid squares with suitable ice thickness could be found throughout almost the entire grids prepared by Preassis. This could be because the increased humidity decreases the water absorption ability of the filter paper. With Preassis, in such a case, the pressure can assist the liquid removal and therefore the humidity has less influence on Preassis than that on Vitrobot. Our results show that Preassis is more efficient in removing viscous liquid and less affected by high humidity compared to Vitrobot.’

We also have added the temperature and humidity information in the Method section, on pages 15 and 16, Supplementary information.

2) The protocol mentions a “Munktell #110067 or similar” but I cannot find specifications online of this particular type of filter paper, except that is of “grade 3”. There are different grades/types of filter paper that one can use with varying results. The type of filter paper is critical, for example, as one with a coarse/loose fiber grain may puncture the grid upon aspiration. Therefore more information on the ideal filter paper specifications would be beneficial in this regard.

Response: We agree with the reviewer that the type of filter paper can also affect the ice thickness in Preassis. In this work, we used Munktell Filtrak™ Grade 3. We have updated this information in the protocol on page 2, Supplementary information:

‘Filter paper (Munktell Filtrak™ Grade3, 55 mm diameter, or similar)’.

3) Microscope parameters such as magnification/pixel size are not present in the section “TEM image collection”, though they (diffraction parameters) are mentioned for the MicroED experiments in the Methods section. Because figs 1-3 and supplementary figs 2-6 have scale bars in the images, then the magnification/pixel size parameters are known, and should ideally be included in the Methods section.

Response: The parameters of the magnifications and pixel sizes have been added in the Methods section. We have merged the following two sections in the Methods in the previous manuscript:

‘TEM image collection. TEM images were collected on a JEOL JEM-2100LaB₆ TEM equipped with an Orius detector. All the images were collected at 200 kV under cryogenic condition using a Gatan 914 cryo-transfer holder. Because of the lens distortion at ultra-low magnification, the images of the grid maps are distorted, especially at the edges.

Electron diffraction data collection. Selected area electron diffraction patterns and MicroED data were collected under cryogenic conditions using a Gatan 914 cryo-transfer holder on a

JEOL JEM-2100LaB₆ TEM operated at 200 kV and recorded by a fast Timepix hybrid pixel detector (Amsterdam Scientific Instruments). The conditions used for the data collections were: spot size 3, camera length 80 cm / 100 cm, and exposure time 1 s / 2 s per frame. MicroED data were collected by continuously rotating the crystal whilst ED frames were simultaneously recorded. The rotation speed of the goniometer was 0.45 °/s.'

into a new section in the current manuscript as shown below (Page 17):

'TEM imaging and ED data collection. TEM images (except for those in **Fig. 2** and **Supplementary Fig. 5**) and ED patterns were collected under cryogenic condition on a JEOL JEM-2100LaB₆ TEM (200 kV) using a Gatan 914 cryo-transfer holder. All the TEM images were taken at the image mode on a Gatan Orius camera (2048 × 2048). An ultra-low magnification (50 ×) was used to image the entire grid. To image grid squares, a magnification range of 100 - 300 × and a pixel size range of 0.15 - 0.9 μm/pixel were used. To image crystals within a grid square, a magnification range of 12000 - 2500 × and a pixel size range of 6.6 - 32 nm/pixel were used. Selected area ED patterns and MicroED data of R2lox (**Supplementary video**) were recorded by a fast Timepix hybrid pixel detector (512 × 512, Amsterdam Scientific Instruments). The conditions used for the data collections were: spot size 3, camera length 80 cm / 100 cm, and exposure time 1 s / 2 s per frame. MicroED data of R2lox were collected by continuously rotating the crystal whilst ED frames were simultaneously recorded. The rotation speed of the goniometer was 0.45 °s⁻¹. The dose rate was estimated to be 0.10 e⁻Å⁻²s⁻¹. The software used for data collection was *Instamatic*²⁹.

TEM images in **Fig. 2** and **Supplementary Fig. 5** and MicroED data in **Supplementary Tables 1** and **2** were collected on a Themis Z microscope (300 kV) equipped with a monochromator and a Gatan OneView camera (4096 × 4096). A Gatan ElsaTM 698 cryo-transfer holder was used to keep the grids at cryogenic condition. Different magnifications were used to image the entire grid (100 ×, 339.0 nm/pixel, 1024 × 1024 with 4× binning) and crystals within a grid square (660 ×, 13.0 nm/pixel, 4096 × 4096). Atlas images (left column of **Supplementary Fig. 5**) were obtained by stitching 36 images (magnification 100 ×) using an in-house script. MicroED data were collected by *InsteaDMatic*³⁰ on the Gatan OneView camera using the *in situ* data capture mode (1024 × 1024, 4× binning). The parameters used for MicroED data collection were: spot size 11, Mono -50, camera length 2.3 m, dose rate 0.03 e⁻Å⁻²s⁻¹, rotation speed 0.57 °s⁻¹, and exposure time 2 s/frame.'

4) It appears that the very low-mag images (in supplementary figs 3-6) of the entire TEM grid were taken in diffraction mode with the diffraction focus set to spread the beam from the focused (condensed) beam, hence the pincushion distortion effect. In this case, the scale bar is highly inaccurate across the image due to extreme distortion, and is better left out. For the intermediate low-mag images (e.g., supplementary fig 3d-f), where the distortion is minimal, the scale bar is applicable because it is generally accurate. Either the cause of the distortion should be mentioned in the text, or preferably, these images should be taken in an imaging lens mode (i.e., "LOW MAG" mode on the JEOL), where the pixel size is known and where there is minimal distortion in the image to render an accurate scale bar.

Response: We want to point out that these images were taken under 'LOW MAG' image mode. The distortions were caused by the lenses on a JEOL JEM-2100LaB₆ microscope. Accordingly, we have removed the scale bars on the LOW MAG images. We have added the following sentences in the corresponding figure captions (Supplementary Figs. 4, 6, 8, 9): 'We note that

the images are distorted at low magnification especially at the edges, resulted from the geometrical distortion of lenses. The periodicity of the grid squares is 86 μm .'

Reviewer #3 (Remarks to the Author)

The manuscript by Zhao and coworkers describes Preassis, a method to prepare crystalline samples for micro-crystal electron diffraction (MicroED) measurements. Sample preparation has long been one of the most finicky aspects of MicroED as evidenced by several publications on the subject, many of which are cited in the manuscript. The difficulty lies in removing just the right amount of buffer around the sample before freezing it and doing so without damaging the crystal lattice. If the ice around the crystals is too thin, the sample may run dry during subsequent data collection; if it is too thick, the beam may not be able to penetrate the sample or the weak high resolution reflections may disappear in the noise contribution of the solvent. If the lattice is damaged, the diffractive power of the crystal will be diminished.

The strength of Preassis lies in its simplicity: it can be implemented using readily available laboratory components. The method is mostly controlled by the user's choice of grid, which determines the size and distribution of holes, and the pressure applied to the backside of the grid. The idea is to tune these two variables such that excess solvent is removed once the sample is dispensed on the grid but crystals remain intact. Based on their success with crystals of two forms of lysozyme, GTPase, ZSM-5, and R2lox crystallized in a viscous buffer, the authors conclude that Preassis is applicable to a wide range of protein crystals and buffer conditions. The case of R2lox is particularly compelling, since it had previously resisted successful structure solution due to a thick, impenetrable solvent layer that could not be adequately blotted owing to its viscosity. The thick solvent would extinguish the weak signal from the faint spots during data collection, leading to a marked decrease in resolution.

The manuscript is well-written and clearly describes the method to the reader. The work on Preassis is original, and the method appears to offer clear advantages to the commonly used blotting and plunging techniques of sample preparation. In the cases shown, Preassis handles viscous solvent more gracefully and was found to preserve tenfold more crystals on the prepared grids than the ubiquitous Vitrobot and it can be implemented at the fraction of the price.

Major comments

Owing to the strong interaction between the electrons and the sample, MicroED excels at obtaining useful diffraction data from minuscule crystals but struggles when crystals are too big. Since finding crystallization conditions that yield smaller crystals is often unfeasible, this is probably the biggest problem in sample preparation for MicroED. The literature contains methods to address this problem by fragmenting crystals or using a focused ion beam to mill crystals into thin sheets. Neither of these techniques are optimal, as one can be quite blunt and the other is often very time-consuming. Sadly, it is also an issue that the manuscript does not address at all, and this dampens the enthusiasm for the method and makes statements about the universality of the method appear somewhat inflated. The authors may assume that the large-crystal problem is solved either pre- or post-Preassis, in which case a brief discussion of this problem should be included in the text.

Response: We agree with the reviewer that finding crystallization conditions that yield microcrystals suitable for MicroED can be a challenging problem and also needs more research. However, there are many cases where microcrystals suitable for MicroED are formed during the crystallization experiments, and these crystals can be studied by MicroED. If the bottle-neck for MicroED specimen preparation can be overcome by Preassis, more novel protein structures can be solved by MicroED. To address this problem, we have added a brief discussion at the end of the manuscript (page 14) as given below:

‘While mechanical crystal segmentation¹⁰ or cryo-FIB milling¹¹⁻¹⁴ can be applied to reduce the size of crystals too large for MicroED, these methods are not optimal. More research is needed to find optimal conditions to directly grow small microcrystals. Furthermore, it is also important to develop new methods for screening such microcrystals because they are hardly visible under light microscopes. We believe, by overcoming the bottle-neck for MicroED specimen preparation, more protein structures can be studied by MicroED.’

Other comments

The manuscript rightly points out that no two samples behave identically, and that each new sample will require parameters individually tuned to the size and shape of the crystals and the viscosity of the solvent around it. For conventional sample preparation with a plunger, this optimization can be quite costly in terms of both time and sample consumption. Even though the lysozyme example makes the relationship between the parameter values and the outcome of Preassis appear very straightforward, the ensuing two-dimensional search space is still substantial. Some information on how much time one may expect to spend on optimizing sample preparation parameters would not be amiss.

Response: We agree with the reviewer’s comment. We have added the following text in the Supplementary information (Page 4), at the end of the Supplementary protocol: ‘A few trials are often needed to find the suitable specimen preparation conditions for a new protein crystal sample, which takes a few hours including specimen preparation and grid screening on a TEM using a cryo-transfer holder.’

Further information about the difficulty of parameter tuning is particularly relevant to the claims of future automation, where this kind of optimization would have to be carried out without human intervention. None of the sample preparation methods currently described in the literature are particularly amenable to automation, and it is not immediately obvious what form such an implementation could take; a more detailed speculation on this topic would be appreciated.

Response: We have added the following text in Supplementary information (Page 5), at the end of the Supplementary protocol: ‘Furthermore, a vertical setup of Preassis (the grid is held vertically and a suction tube is placed behind the grid) can be implemented as an add-on to the Vitrobot to enable environmental control and automated plunge-freezing. Preassis can be also applied to pre-clipped EM grids used for auto-loading, which makes this method very promising for future automation.’

The amount of time the sample is left on the filter paper is another parameter in Preassis. Even

though it varies by a factor of two for the examples given, it is not discussed much in the text. Presumably, the proposed enhancement of using a humidity chamber is related to this time, so it must have some impact on the result. A related question concerns the speed at which the grid is plunged into liquid ethane. Are there any effects of uneven vitrification, since grids in Preassis are plunged manually?

Response: We agree with the reviewer that time is also another parameter in Preassis. We have added the following text in the Supplementary protocol (page 4, supplementary information) to address this: ‘The suction time may also affect the thickness of the vitrified ice. In this work, the suction time was kept at ~ 5 s for non-viscous crystal suspensions and ~10 s for viscous crystal suspensions. Based on the current setup of Preassis, the time is less controllable (due to manual grid handling) compared to the other two parameters pressure and hole size of the carbon film. A detailed study will be performed when a more controllable and automated setup is built.’

We didn’t see uneven vitrification due to the manually plunge-freezing. Instead, we found crystalline ice when the ice layer was too thick, which is a common phenomenon in specimen preparation using both Vitrobot and Preassis. The speed of plunge-freezing could affect the vitrification when the crystal suspension doesn’t include cryo-protectants (e.g PEG) or a relatively high concentration of salts. We have added the following text in the Supplementary protocol at step 6, on page 4 of the Supplementary information: ‘The manual plunging-freezing could affect the freezing speed and therefore the vitrification. If the original crystal suspension does not contain any cryo-protectants (e.g high molecular weight polymers) or has a relatively low salt concentration, it may be necessary to add a suitable cryo-protectant to minimize the chance for crystalline ice formation.’

Minor comments and suggested improvements.

Were crystals ever observed to break due to the applied pressure? Presumably, this does happen, as the tetragonal lysozyme crystals shown in the manuscript rarely extend into the grid holes, even when they are sitting close their edges.

Response: We didn’t notice any break of crystals due to the applied pressure. This is confirmed by both the TEM images and the quality of MicroED data (Supplementary Table 1 and Table 2) collected from the grids prepared by Vitrobot and Preassis. The reason why most of the tetragonal lysozyme crystals rarely lay on top of the grid holes could be explained by that the average size ($< 1 \mu\text{m}$) of crystals is smaller than the hole size ($1.2 \mu\text{m}$) of the holey carbon film. Therefore, most of the submicron-sized crystals can either stay on top or near the edge of the holey carbon film or go through the holes. When the diameters of the crystals (e.g R2lox and GTPase) are larger than that of the hole size, we can find a lot of crystals laying on top of the holes as shown in Fig. 3 and Fig. 5b-c.

Depending on what constitutes a protein versus a peptide, there are arguably more than 16 unique MicroED structures in the PDB as of this writing. Since this number is rapidly changing, it may be better to give a more approximate count in the text.

Response: We have changed the sentence on page 2 to ‘However, until now, only a few macromolecular structures solved by MicroED are reported in the PDB database and all except R2lox³ had been previously determined by X-ray diffraction.’

REVIEWER COMMENTS

Reviewer #1 (Remarks to the Author):

The authors have addressed all of my major concerns on the manuscript

Reviewer #2 (Remarks to the Author):

The revised manuscript is acceptable. Please note the following typographical errors in the corrected text:

- main text, page 17, line 347: "MciroED" should be "MicroED"
- supplementary text, page 10, line 196: "Arius" should be "Orius"

Reviewer #3 (Remarks to the Author):

All previous issues were satisfactorily addressed. Minor comments:

The table header and the legend for figure 2 disagree: the header for the third column reads "Vitrobot", whereas the legend says "Preassis" for panels g-i.

Line 347: "MciroED" -> "MicroED"

MS No: NCOMMS-21-03438A

Response letter to the Reviewers's comments on "A simple pressure-assisted method for MicroED specimen preparation" by Zhao et al.

We would like to give our sincere thanks to the Reviwers for the valuable comments. Please find the point-to-point responses below, which are given in blue. The changes and new text in the revised manuscript are marked in yellow.

Reviewer #1 (Remarks to the Author):

The authors have addressed all of my major concerns on the manuscript.

Reviewer #2 (Remarks to the Author):

The revised manuscript is acceptable. Please note the following typographical errors in the corrected text:

- main text, page 17, line 347: "MciroED" should be "MicroED"
- supplementary text, page 10, line 196: "Arius" should be "Orius"

Response:

We have made corresponding changes in the main text and supplementary text.

Reviewer #3 (Remarks to the Author):

All previous issues were satisfactorily addressed. Minor comments:

The table header and the legend for figure 2 disagree: the header for the third column reads "Vitrobot", whereas the legend says "Preassis" for panels g-i.

Line 347: "MciroED" -> "MicroED"

Response:

We have made corresponding changes in the main text.